



**Measurement report: Ambient volatile organic compounds (VOCs) pollution at urban**
**Beijing: characteristics, sources, and implications for pollution control**
Lulu Cui[1], Di Wu[1], Shuxiao Wang[1,2*], Qingcheng Xu[1], Ruolan Hu[1], Jiming Hao[1, 2]
*[1] State Key Joint Laboratory of Environment Simulation and Pollution Control, School of Environment,*
*Tsinghua University, Beijing 100084, China*
*[2] State Environmental Protection Key Laboratory of Sources and Control of Air Pollution Complex, Beijing*
*100084, China*
* Corresponding author. E-mail address: shxwang@tsinghua.edu.cn
## Abstract
The increasing ozone ($O_3$) pollution and high fraction of secondary organic aerosols (SOA) in fine particle mass
highlighted the importance of volatile organic compounds (VOCs) in air pollution control. In this work, a
campaign of comprehensive field observations was conducted at an urban site in Beijing, from December 2018
to November 2019, to identify the composition, sources, and secondary transformation potential of VOCs. The
total mixing ratio of the 95 quantified VOCs (TVOC) observed in this study ranged from 5.5–118.7 ppbv with
the mean value of 34.9 ppbv, and the contemporaneous mixing ratios of TVOC was significantly lower than those
observed in 2014 and 2016, confirming the effectiveness of VOCs emission control measures in Beijing in recent
years. Alkanes, OVOCs and halocarbons were the dominant chemical groups, accounting for 75-81% of the
TVOCs across the sampling months. High and low-$O_3$/$PM_{2.5}$ months as well as several $O_3$/$PM_{2.5}$ polluted days
were identified during the sampling period. By deweathered calculation, we found that high $O_3$/$PM_{2.5}$ levels
were due to both enhanced precursor emission levels and meteorological conditions favorable to
$O_3$ and $PM_{2.5}$ production. The molar ratios of VOCs to $NO_X$ indicated that $O_3$ formation was limited by VOCs
during the whole sampling period. Diesel exhaust and industrial emission were identified as the major VOCs





sources on both $O_3$-polluted and $PM_{2.5}$-polluted days based on positive matrix factorization (PMF) analysis,
accounting for 46% and 53%, respectively. Moreover, higher proportion of oil/gas evaporation was observed on
$O_3$-polluted days (18%) than that on $O_3$-clean days (13%), and higher proportion of coal/biomass combustion
was observed on $PM_{2.5}$-polluted days (18%) than that on $PM_{2.5}$-clean days (13%). On the base of $O_3$ formation
impact, VOCs from fuel evaporation and diesel exhaust particularly toluene, xylenes, trans-2-butene, acrolein,
methyl methacrylate, vinyl acetate, 1-butene and 1-hexene were the main contributors, illustrating the necessity
of conducting emission controls on these pollution sources and species for alleviating $O_3$ pollution. Instead, VOCs
from diesel exhaust and coal/biomass combustion were found to be the dominant contributors for secondary
organic aerosol formation potential (SOAFP), particularly the VOC species of toluene, 1-hexene, xylenes,
ethylbenzene and styrene, and top priority should be given to these for the alleviation of haze pollution. The
positive matrix factorization (PSCF) analysis showed that $O_3$ and $PM_{2.5}$ pollution was mainly affected by local
emissions.  This study provides insights for government to formulate effective VOCs control measures for air
pollution in Beijing.
**Key words:**  VOCs, OFP, SOAFP, Source appointment



## 1. Introduction

The ozone (O₃) and fine particulate matter (PM₂.₅) pollution has restricted improvements in air quality in China.

Observation data from the Chinese Ministry of Environment and Ecolgy (MEE) network has witnessed an upward

trend for O₃ across the country over the period 2013-2019 (Fu et al., 2019; Li et al., 2017; Li et al., 2020; Shen

et al., 2019; Fan et al., 2020). Besides, haze pollution occurred in urban sites were commonly characterized by

high fractions of secondary organic aerosols (SOA) in fine particles (Guo et al., 2014; Huang et al., 2014). Volatile

organic compounds (VOCs) are key precursors for the formation of O₃ via multiphase reactions (Odum et al.,

1997; Atkinson, 2000; Sato et al., 2010; Huang et al., 2014). In highly polluted urban regions, the O₃ formation

was generally VOCs-limited, and it is suggested that VOCs emission control is necessary for effective alleviation

of photochemical smog (Liu et al., 2020a,b; Shao et al., 2009; Wang et al., 2020; Xing et al., 2011). Besides, the

VOCs compounds including aromatics and biogenic species have significant impact on SOA formation which

play an important role in haze formation (Hallquist et al., 2009; Huang et al., 2014). VOCs emission abatement

is therefore imperative for improving air quality in China.

VOCs in ambient air can be emitted by a variety of sources including both anthropogenic and biogenic

sources. While biogenic emissions are more than 10 times that of anthropogenic emissions globally (Roger and

Janet, 2003), anthropogenic emissions play the dominant role in urban and surrounding areas (Warneke et al.,

2007; Ahmad et al., 2017; Wu and Xie, 2018). The VOC observations in China showed distinct differences in

anthropogenic sources among different regions. For example, solvent use and vehicle exhaust are primary VOCs

sources in urban Shanghai and urban Guangzhou, while the primary sources of VOCs in Wuhan, Zhengzhou and

Beijing cities are combustion and vehicle exhaust (Han et al., 2020; Shen et al., 2020; Liu et al., 2020a; Li et al.,

2019a). Apart from the diversity of emission sources, different VOCs species exhibited different propensities to

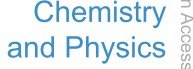

form $O_3$ and SOA. Observation-based studies commonly applied the $O_3$ formation potential (OFP) and SOA
formation potential (SOAFP) scales to quantify the relative effects of specific VOCs and sources on $O_3$ and SOA
formation and to aid in the development of efficient control strategies (Carter and Atkinson, 1989; Chang and
Rudy, 1990; Han et al., 2020; Zhang et al., 2017a).  Although there have been many studies on ambient VOCs in
various locations (e.g., urban, rural, and industrial areas), most of these measurements were confined to short
periods (a few days or a certain season), and the understanding of temporal variations of concentrations, sources
as well as the influence of photochemical reactions of VOCs on annual scale was still limited. Besides, most of
the available reports on VOCs analysis based on online analytical techniques include mainly non-methane
hydrocarbon compounds, and thus the characteristics of VOCs as well as their relationships with $PM_{2.5}$ and $O_3$
cannot be fully revealed since OVOC also participate actively in chemical reactions related to secondary
formation (Li et al., 2019a; Zhao et al., 2020; Yang et al., 2018; Sinha and Sinha., 2019). Therefore, the long-
term and comprehensive monitoring of VOCs are desired.

70        As the capital and one of the largest megacities in China, Beijing has been suffering from severe $O_3$ pollution

due to rapid economic development and increases in precursor emissions (Wang et al., 2014; Wang et al, 2017;
Li et al., 2019d; Zhao et al., 2020). According to the Report on the State of the Ecology and Environment in
Beijing, the average 90th percentile $O_3$ daily maximum 8 h concentration in Beijing exceeded the national
standards, reaching 193, 192, and 191 µg/m$^3$ in 2017, 2018, and 2019, respectively. In addition, the number of
motor vehicles in Beijing reached 6.365 million at of the end of 2019 (http:// beijing.gov.cn), making Beijing the
top city in China in terms of number of motor vehicles. The existing field measurements in Beijing were mostly
conducted before 2016, and the observation in most recent years is quite limited (Li et al., 2015; Li et al., 2019c;
Liu et al., 2020a; Yang et al., 2018). In this work, a campaign of comprehensive field observations was conducted


at an urban site in Beijing during December 2018 and November 2019 for the analysis of VOCs. Several $O_3$ and
$PM_{2.5}$ pollution events were captured during the sampling period. The characteristics and the contribution of
specific species and sources of VOCs on $O_3$ and SOA formation, with a focus on photochemical and haze
pollution periods, were analyzed in detail. The results and implications from this study can provide useful guidance
for policymakers to alleviate ozone and haze pollution in Beijing.

## 84  2. Methodology

### 85  2.1 Field measurement

The sampling site is at the roof of a three-floor building on the campus of Tsinghua University (40.00°N,
116.33°E), northwest of Beijing urban area (Fig. S1). The altitude of the sampling site is 57 m. This sampling
site is surrounded by school and there are no large emission sources nearby, therefore it can represent the urban
air quality in Beijing. Details of the site description is found in Xu et al., (2019).
The air samples were collected using 6 L summa canisters (Entech, USA) with a stable rate of 4.26 ml/min.
The samples were pre-processed to remove $N_2$, $O_2$, $CO_2$, CO and $H_2O$ in the samples and to further concentrate
the samples in volume by the cryogenic pre-concentrator (Model 7100, Entech Instruments Inc., USA).  Pressure
gage was used to test if the canister has air leakage exist before sampling every time, and blanks were prepared
using cleaned canisters to fill with high purity nitrogen. The cryotraps of precooling system was baked before
analyses each day and between every samples. The VOCs in air samples were analyzed by a gas chromatography
system (Agilent Tech., 7890/5975, USA). The column temperature was controlled by an initial temperature of -
40 °C. The programmed temperature was used with helium as carrier gas, and the flow rate was set at 1.5 ml min⁻
¹. The initial temperature was set at 90 °C, and then switched to 220 °C. The standard substance (SPECTRA
GASES Inc., USA) mentioned for Photochemical Assessment Monitoring Stations (PAMS) and US EPA TO-15



standard was used to construct the calibration curves for the 95 target VOCs, including 25 alkanes, 8 alkenes, 16
aromatics, 34 halocarbons and 12 OVOC. Quality assurance and quality control, including method detection limit
(MDL) of each compound, laboratory and field blanks, retention time, accuracy and duplicate measurements of
samples were performed according to USEPA Compendium Method TO-15 (USEPA 1999). The correlated
coefficient of the calibration curves for all the compounds was > 0.95. The relative standard deviation for all of
compounds of triplicates were 0.5%-6.0%

106        During the sampling periods, the measurements of $PM_{2.5}$, gaseous pollutants ($NO_X$ and $O_3$), and

meteorological variables (such as temperature, relative humidity, wind speed, and wind direction) were
conducted simultaneously. $NO_X$ and $O_3$ were analyzed using the Ozone Analyzer (Thermo Fisher Scientific USA,
49I) and $NO$–$NO_2$-$NO_X$ Analyzer (Thermo Fisher Scientific USA, 17I), respectively. The mass concentration of
$PM_{2.5}$ was measured using an oscillating balance analyzer (TH-2000Z, China) (Wang et al., 2014).
Meteorological including wind speed (WS), wind direction (WD), relative humidity (RH), air pressure,
temperature, air pressure, and precipitation were measured by an automatic weather monitoring system. The
planetary boundary height was obtained from the European Centre for Medium-Range Weather Forecasts
(https://www.ecmwf.int/en/forestcasts/datasets/browse-reanalysis -datasets).
**2.2 Ozone formation potential (OFP) and secondary formation potential (SOAFP) calculation**
The formation potential of $O_3$ and SOA was used to characterize the relative importance of VOCs species and
sources in secondary formation, which were estimated using Eqs. (1) and (2).
$$OFP = \sum_{i}^{n} MIR_i \times [VOC(ppb)]_i \tag{1}$$

$$SOAFP = \sum_{i}^{n} Y_i \times [VOC(ppb)]_i \tag{2}$$



where $n$ represents the number of VOCs, [VOC]$i$ represents the $i$th VOC species concentration, MIR$i$ is the
maximum incremental reactivity for the $i$th VOC species, and $Yi$ is the SOA yield of VOC$_i$ (McDonald et al.,
2018). The MIR for each VOC species were taken from the updated Carter research results
(http://www.engr.ucr.edu/~carter/reactdat.htm, last access: 24 February 2021). For species lacking yield curves,
the fractional aerosol coefficient (FAC) values proposed by Grosjean and Seinfeld (1989) were used.

**2.3 Deweathered model**

In this work, a random forest (RF) model was used to assess the meteorology-associated variations and quantify
the impacts of precursor emissions to O$_3$ and PM$_{2.5}$ levels. The meteorological predictors in the RF model include
wind speed (*WS*), wind direction (*WD*), air temperature (*T*), relative humidity (*RH*), precipitation (*Prec*), air
pressure (*P*), time predictors (year, day of year (*DOY*), hour) and planetary boundary layer height (*BLH*). These
meteorological parameters have been reported to be strongly associated with PM$_{2.5}$ and O$_3$ concentrations in
various regions in China (Chen et al., 2020; Feng et al., 2020) and contributed significantly in previous PM$_{2.5}$
and O$_3$ prediction models (She et al., 2020; Li et al., 2020). The original dataset was randomly classified into a
training dataset (90 % of input dataset) for developing the RF model, and the remaining one was treated as the
test dataset. After the building of the RF model, the deweathered technique was applied to predict the air pollutant
level at a specific time point. The differences in original pollutant concentrations and deweathered pollutant
concentrations were regarded as the concentrations contributed by meteorology. Statistical indicators including
$R^2$, RMSE, and MAE values were regarded as the major criteria to evaluate the modeling performance.

**2.4 Positive matrix factorization (PMF)**

In this study, the US EPA PMF 5.0 software was used for VOCs source apportionment (Abeleira et al., 2017; Li
et al., 2019a; Xue et al., 2017). The detailed description of the PMF model is found elsewhere (Ling et al., 2011;



Yuan et al., 2009). PMF uses both concentration and user-provided uncertainty associated with the data to weight
individual points. Species with high percentages of missing values (> 40 %) and with signal-to-noise ratio of
below 2 were excluded. Based on this, 53 VOC species including source tracers (e.g., chloromethane,
trichloroethylene, tetrachloroethylene and MTBE) and $SO_2$ were chosen for the source apportionment analysis.
Data values below the MDL were replaced by MDL/2, and the missing data were substituted with median
concentrations. If the concentration is less than or equal to the MDL provided, the uncertainty is calculated using
the equation of Unc = 5/6 × MDL; if the concentration is greater than the MDL provided, the uncertainty is
calculated as Unc = [(error faction × mixing ratio)$^2$+ (MDL)$^2$]1/2.
**2.5 Cluster and potential source contribution function (PSCF) analysis**
The potential source contribution function (PSCF) model has been widely used to identify potential source
regions of air pollutants (Hong et al., 2019; Liu et al., 2016b, 2019, 2020a; Zheng et al., 2018;). In this study, the
24 h backward trajectories (1 h intervals) of air masses arriving at the sampling site with a trajectory height of
1500 m were calculated using the MeteoInfoMap software. The study area covered by back trajectories was
divided into 0.5°×0.5° grid cells. The pollution trajectory was defined as the trajectories corresponding to the
total VOC (TVOC) concentration that exceeded the 75th percentile concentration of TVOC. PSCF value in the
*ij* th grid was defined as:
$$PSCF_{ij} = \frac{m_{ij}}{n_{ij}}$$   (3)
where the $m_{ij}$ is the number of polluted trajectories through the grid; $n_{ij}$ is the all trajectories through the grid.
The weight function $W_{ij}$ is applied to reveal the uncertainty of small values of nij (Polissar et al., 1999; Li et al.,

160   2017):



$$W_{ij} = \begin{cases} 1.00 & 80 < n_{ij} \\ 0.70 & 20 < n_{ij} \leq 80 \\ 0.42 & 10 < n_{ij} \leq 20 \\ 0.05 & n_{ij} \leq 10 \end{cases}$$
(4)

## 3. Results and discussion

### 3.1 TVOC mixing ratios and chemical composition

The time series of meteorological parameters and concentrations of air pollutants during the measurement
period are shown in Fig. 1. The ambient temperature ranged from -13.3°C to 38.7°C and the RH varied between
5% and 99% across the sampling months. Prevailing winds shifted between southwesterly and northeasterly with
WS of 0–6.8 m s$^{-1}$. The mixing ratio of total VOCs (TVOC) ranged from 5.5–118.7 ppbv during the sampling
period with relatively higher values during September and November (49.9-51.6 ppbv) while relatively lower
values (22.2-27.5 ppbv) across the other months. Major VOC compositions were generally consistent during the
whole measurement period. Alkanes, OVOCs and halocarbons were the dominant chemical groups, accounting
for 75-81% of the TVOCs across the sampling months. In terms of individual species, acetone, dichloromethane,
butane, toluene, methyl tert butyl ether (MTBE), *i*-pentane, propylene, hexane, 1,1- dichloroethane, benzene and
1-butene made up the largest contribution, accounting for 50.6 % of the TVOC on average during the whole
measurement period.
The comparison of concentration and composition of chemical groups observed in this work and previous
studies is shown in Fig. 2. Clearly, the concentrations of TVOCs and major VOC groups observed in this study
were apparently lower than those in 2014 and 2016 in urban sites in Beijing (An et al., 2012; Liu et al., 2020a;
Li et al., 2015b), indicating the effectiveness of control measures in most recent years on lowering VOCs emission.
Besides, the composition of major chemical groups also showed remarkable changes, with decreased proportions





of alkanes while increased fractions of halocarbons, aromatics and OVOCs, reflecting the changes in emission
sources types in most recent years.

182        During the measurement period, nine $O_3$ pollution events (maximum daily 8-h average $O_3$ exceeding 160

$\mu g\ m^{-3}$) were observed, which occured during 17-22 April, 3-17 May, 18-29 June, 2-13 July, and 25-29 September
of 2019. The months with $O_3$ pollution events were classified as high-$O_3$ months in this study, which were further
classified into the clean and polluted days based on the measured concentrations. During the four high-$O_3$ months
(i.e., April, May, June, July, and September), the WS on polluted days ($1.31 \pm 0.90$ m s$^{-1}$) was slightly lower than
that on clean days ($1.47 \pm 1.10$ m s$^{-1}$), indicating that precursors were more conductive to be diluted on clean
days. The variation trend of $O_3$ and temperature displayed the negative correlation, and the linear correlations
between $O_3$ and temperature on polluted days ($R^2 = 0.63$) was stronger than that on clean days ($R^2 = 0.35$). The
comparison of meteorological parameters and air pollutants concentrations on clean and polluted days is shown
in Fig. 3. The mean TVOC concentration showed relatively higher values on pollution days (32.3 ppbv) than that
on clean days (29.6 ppbv), which was mainly contributed by higher concentrations of MTBE, acrolein, trans-2-
butene was higher on polluted days. MTBE is widely used as a fuel additive in motor gasoline (Liang et al., 2020),
and trans-2-butene is the main component of oil/gas evaporation (Li et al., 2019a). Such result indicated enhanced
contribution of traffic emissions on polluted days. Besides, the concentration of isoprene, which is primarily
produced by vegetation through photosynthesis, increased significantly during the $O_3$ pollution day probably due
to the stronger plant emission at elevated temperature (Guenther et al., 1993, 2012; Stavrakou et al., 2014). The
ratio of $m/p$-xylene to ethylbenzene (X/E) measured can be used as an indicator of the photochemical aging of
air masses because of their similar sources in urban environments and differences in atmospheric lifetimes
(Carter., 2010; Miller et al., 2012; Wang et al., 2013). The mean X/E value on $O_3$ clean days (1.41) was higher



than that on polluted days (1.17), indicating enhanced secondary transformation of VOCs to $O_3$ during the
polluted periods.
The daily $PM_{2.5}$ concentrations ranged from 9-260 µg m$^{-3}$ with the mean value of 88.5 µg m$^{-3}$ during the
measurement period. Fourteen $PM_{2.5}$ pollution events (daily average $PM_{2.5}$ exceeding 75 µg m$^{-3}$) were observed,
which occured on 1-2 December and 5 December of 2018, 3 January, 12-13 January, 22-23 April, 29 April, 12
May, 15 May, 19 October, and 21-23 November of 2019.  The months with $PM_{2.5}$ pollution events were classified
as high-$PM_{2.5}$ months, which were also further classified into the clean and polluted days based on the measured
concentrations. The WS on polluted days (1.05 ± 1.06 m s$^{-1}$) was lower than that on clean days (1.43 ± 1.06 m s$^{-}$
$^1$), indicating the weaker ability of winds for the dilution and diffusion of precursor on polluted days. Both the
value of relative humidity (RH) and TVOCs increased significantly on polluted days, suggesting that the
secondary transformation of VOCs was more conducive at high RH. The mean X/E value on $PM_{2.5}$ clean days
(1.47) was slightly higher than that on polluted days (1.44), indicating enhanced secondary transformation of
VOCs to $PM_{2.5}$ during the pollution periods.
**3.2 The role of VOCs on secondary pollution**
**3.2.1 Estimating $O_3$ and $PM_{2.5}$ levels contributed by emissions**
$O_3$ and secondary aerosols are primarily formed via photochemical reactions in the atmosphere, of which
concentrations could be largely influenced by meteorological conditions (Chen et al., 2020; Feng et al., 2020;
Zhai et al., 2019). In this work, the respective contributions of meteorology and emissions to $PM_{2.5}$ and $O_3$
variations were determined using the RF model as described in section 2.3. The coefficients of determination ($R^2$)
for the RF model in predicting $PM_{2.5}$ and $O_3$ are 0.85 and 0.91, respectively (Shown in Fig. S2). The respective
contributions of anthropogenic and meteorology to $O_3$ and $PM_{2.5}$ during each period is shown in Fig. 4. During



222 the high-$O_3$ months, the meteorologically-driven $O_3$ on the polluted days (72.5 µg m$^{-3}$) was significantly higher

223 than that on the clean days (35.3 µg m$^{-3}$). After removing the meteorological contribution, the residual emission-

224 driven $O_3$ on polluted days (45.3 µg m$^{-3}$) and clean days (44.9 µg m$^{-3}$) of the high-$O_3$ months was almost identical

225 and were significantly higher than that during the low-$O_3$ months (23.8 µg m$^{-3}$). The emission-driven PM$_{2.5}$ level

226 was in the order of: polluted days of the high-PM$_{2.5}$ months (55 µg m$^{-3}$) > clean days of the high-PM$_{2.5}$ months

227 (44 µg m$^{-3}$) > low-PM$_{2.5}$ months (29 µg m$^{-3}$). These results suggested that apart from meteorological factors,

228 emissions also play a role in deteriorating PM$_{2.5}$ and $O_3$ pollution, and reducing anthropogenic emissions is

229 essential for improving air quality.

230  The VOCs/NO$_X$ ratio has been widely used to distinguish whether the $O_3$ formation is VOC limited or NO$_X$

231 limited (Li et al., 2019a). Generally, VOC-sensitive regime occurs when VOCs/NO$_X$ ratios are below 10 while

232 NO$_X$-sensitive regime occurs when VOCs/NO$_X$ ratios are higher than 20 (Hanna et al., 1996; Sillman, 1999). In

233 this study, the values of VOCs/NO$_X$ (ppbv ppbv$^{-1}$) were all below 3 during both the $O_3$-polluted and low-$O_3$

234 months (Fig. S3), suggesting that the $O_3$ formation was sensitive to VOCs, and thus the reductions of the

235 emissions of VOCs will be beneficial for $O_3$ alleviation.

236 **3.2.2 Contribution of VOCs to OFP and SOAFP**

237  As discussed in 3.1, $O_3$ formation was generally VOCs-sensitive during the measurement period.

238 Quantifying the contribution of speciated VOCs species to $O_3$ is helpful for developing effective VOCs control

239 measures and alleviating $O_3$ pollution. The averaged OFP on clean days of the high-$O_3$ months, polluted days of

240 the high-$O_3$ months, and during the low-$O_3$ months were 201.4, 224.9 and 187.5 µg m$^{-3}$, respectively (Fig. 5).

241 According to our observations, the higher OFP on $O_3$-polluted days of the high-$O_3$ months compared with that

242 on clean days of the high-$O_3$ months was mainly contributed by higher levels of trans-2-butene, o-xylene and



acrolein on polluted days, in line with that in Fig. 3. Alkenes, aromatics and OVOCs were the three contributing
chemical groups to $O_3$ formation, accounting for 85.7%, 85.1% and 81.6% of the total OFP on clean days of the
high-$O_3$ months, polluted days of the high-$O_3$ months, and during the low-$O_3$ months, respectively. In terms of
the individual species, the top 10 highest contributors during the high-$O_3$ months were toluene (7.5% and 6.4%
on $O_3$ clean and polluted days, respectively), trans-2-butene (7.5% and 9.6%), acrolein (5.7% and 10.8%), m/p-
xylene (6.9% and 6.1%), o-xylene (5.8% and 6.6%), 1-butene (7.1% and 5.2%), 1-hexene (5.4% and 4.4%), vinyl
acetate (5.7% and 4.2%), methyl methacrylate (4.8% and 5.5%), and 1-pentene (4.4% and 4.5%). During the
low-$O_3$ months, the overall OFP was mainly contributed by toluene (10.8%), trans-2-butene (10.5%), 1-butene
(7.3%), m/p-xylene (6.5%), 1-pentene (5.7%), 1-hexene (5.0%), methyl methacrylate (4.9%), o-xylene (4.9%),
vinyl acetate (3.8%), and isopentane (2.3%), respectively.
As shown in Fig. 4, the ratio of VOCs/NOx was generally below 3 during the sampling period, indicating
high $NO_X$ conditions. Based on the estimated yields of the VOCs shown in Table S1, the SOAFPs were calculated
and compared in Fig. 5. The mean SOAFP on clean days of the high-$PM_{2.5}$ months, polluted days of the high-
$PM_{2.5}$ months, and during the low-$PM_{2.5}$ months were 1.07, 1.28 and 0.89 µg m$^{-3}$, respectively. The higher
SOAFP on $PM_{2.5}$-polluted days of the high-$PM_{2.5}$ months than that on clean days of the high-$PM_{2.5}$ months was
mainly contributed by higher levels of 1,2,4-trimethylbenzene, n-undecanone, n-Nonane, 1,4-diethylbenzene,
and 1,3-diethylbenzene on polluted days. Aromatics have the largest SOAFP, accounting for 75%, 74% and 70%
of the total SOAFP on clean days of the high-$PM_{2.5}$ months, polluted days of the high-$PM_{2.5}$ months and during
the low-$PM_{2.5}$ months, respectively. The 10 species responsible for most of the SOAFP were toluene (41% on
polluted days of the high-$PM_{2.5}$ months, 40% on clean days of the high-$PM_{2.5}$ months, and 33% during the low-
$PM_{2.5}$ months), 1-hexene (13.0%, 12.5%, and 15.2%), xylenes (11.6%, 14.1% and 14.8%), ethylbenzene (4.9%,





5.3% and 6.0%), styrene (4.5%, 5.6% and 5.6%), 1-pentene (3.3%, 3.4% and 4.3%), methyl cyclopentane (2.1%,
2.7% and 3.6%), 1,2,3-trimethylbenzene (2.8%, 2.4% and 2.8%), m-ethyl toluene (1.7%, 1.4% and 1.7%) and p-
ethyl toluene (1.7%, 1.4% and 1.7%), respectively.
**3.3 Source apportionment of VOCs**
The factor profiles given by PMF and the contribution of each source to ambient VOCs during each period
is presented in Fig. 6 and Fig. 7, respectively. Six emission sources were identified: coal/biomass burning, solvent
use, industrial sources, oil gas evaporation, gasoline vehicle emission, and diesel vehicle emission based on the
corresponding markers for each source category. In general, diesel vehicle exhaust, gasoline vehicle exhaust and
industrial emissions were the main VOCs sources during both $O_3$-polluted and $PM_{2.5}$-polluted months, with
contributions of 62%, 62%, 52% and 66% on cleans days of the $O_3$-polluted months, polluted days of the $O_3$-
polluted months, clean days of the $PM_{2.5}$-polluted months, and polluted days of the $PM_{2.5}$-polluted months,
respectively. Diesel and gasoline vehicle exhaust exhibited obvious higher contributions while combustion and
industrial sources showed lower contributions during the high-$O_3$ months than that during the low-$O_3$ months.
The contributions of industrial emissions (22%) on $O_3$-polluted days were much higher than those on clean days
(18%). Besides, the contribution of fuel evaporation (18%) also increased from 13% on $O_3$-clean days to 18% on
$O_3$-pollution days. Figure 8 presents the relative contributions of individual VOC sources from PMF to OFP. On
the base of $O_3$ formation impact, diesel and gasoline vehicle exhaust were major contributors as well, and the
OFP of vehicle emissions on $O_3$-polluted days (93.9 µg m$^{-3}$) was higher than that on $O_3$-clean days (88.0 µg m$^{-
3}$). Besides, fuel evaporation also showed higher OFP (35.5 µg m$^{-3}$) and served as an important contributor (18%)
for $O_3$ formation on $O_3$-polluted days. Although industrial emissions act as an important source for VOCs
concentrations on $O_3$-polluted days, the potential to form $O_3$ is limited, accounting for 11% of the total OFP. As



shown in Fig., the industrial source was distinguished by high compositions of alkanes while relatively lower
compositions of alkenes and aromatics, resulting in low $O_3$ formation potentials. Such results suggested that the
fuel use and diesel vehicle exhaust should be controlled preferentially for $O_3$ mitigation.
The high-PM$_{2.5}$ months showed higher proportions of diesel vehicle emission (18%-24%) while lower
proportions of industrial emission (12%-15%) compared with the low-PM$_{2.5}$ months (12% for diesel vehicle
emission and 31% for industrial emission). The PM$_{2.5}$-polluted days were dominated by industrial emission (29%),
diesel vehicle exhaust (24%), and combustion source (18%). Besides, the contribution of diesel vehicle exhaust
and industrial sources on PM$_{2.5}$-polluted days (24% and 29%) were much higher than those on clean days (18%
and 17%), and the contribution of combustion also increased to 18% during the transition of PM$_{2.5}$-clean days to
18%. On the base of PM$_{2.5}$ formation impact, diesel vehicle exhaust and combustion were two major contributors
on PM$_{2.5}$-polluted days, and these two sources showed obvious higher SOAFP on PM$_{2.5}$-polluted days (0.30 and
0.32 µg m$^{-3}$, respectively) than that on PM$_{2.5}$-clean days (0.15 and 0.14 µg m$^{-3}$, respectively).  Although industrial
emissions act as an important source for VOCs concentrations on PM$_{2.5}$-polluted days, the potential to form PM$_{2.5}$
is limited, accounting for 16% of the total SOAFP. The above results suggested that diesel vehicle exhaust and
combustion should be controlled preferentially for alleviating PM$_{2.5}$ pollution.
Based on the mass concentrations of individual species in each source, m/p-xylene, o-xylene, methyl
methacrylate, vinyl acetate, 1-hexene, and acrolein in gasoline and diesel vehicular emissions; toluene, trans-2-
butene, and 1-pentene in fuel evaporation and diesel vehicular emissions; acrolein in solvent, gasoline vehicular
and diesel vehicular emissions were the dominant species contributing to photochemical $O_3$ formation (Fig. 9).
Toluene, m/p-xylene, o-xylene, styrene, ethylbenzene, 1-pentene, 1,2,3-trimethylbenzene from combustion and
diesel vehicular emissions; 1-hexene from diesel vehicular emission; and methyl cyclopentane from combustion,





industrial and diesel vehicular emissions were the dominant contributors for SOA formation during the $PM_{2.5}$
pollution periods (Fig. 9).

**3.4 Influence of source regions**

Regional transport is an essential source for VOCs in addition to local emissions. The possible geographic origins
of the VOC sources were explored using PSCF as shown in Fig. 10 and 11. Diversities of geographic origins
were found for different periods. During the high-$O_3$ months, high PSCF values were found in Beijing and the
junction of Hebei province while the PSCF showed high values in Inner Mongolia, northern Shanxi, Hebei, and
Beijing during the clean months. The air mass backward trajectory cluster analysis indicated that the VOCs
concentration was significantly affected by pollution transmission at the Shanxi Province and Hebei Province
junction and local source emissions during the high-$O_3$ months. Note that the west short distance trajectories
(cluster 1 in Fig. 10a) that passed over Shanxi and Hebei provinces exhibited relatively high VOCs concentrations
and OFP, mainly contributed by aromatics. In addition, although the VOCs concentration of the trajectories from
the south (cluster 3 in Fig. 10) was relatively small, the OFP in these pollution trajectories were the highest due
to relatively higher proportion of aromatics. Therefore, in addition to local emissions, the transmission of highly
polluting air masses from the western and southern Hebei should be paid attention for controlling emissions of
reactive VOCs compounds. During the low-$O_3$ months, the VOCs concentration was mainly affected by the
northwest trajectory that originate from Inner Mongolia and passed through the western Hebei, reflecting large-
scale and long-distance transport of VOCs. Besides, the air masses from the southern region (cluster 1 in Fig.
10c), representing pollution transmission from Shanxi and Hebei provinces also have significant impact on VOCs
concentration. Note that the VOCs concentration during the $O_3$-clean months were relatively higher compared
with those during the polluted months. Fast-moving air masses from the northwest Inner Mongolia typically carry



clean air masses (Zhang et al., 2017b). It is proposed that the higher VOCs concentration during the low-$O_3$
months was related to domestic coal/biomass burning in the northern regions during cold seasons. However, the
OFP during the low-$O_3$ months were generally lower than those during pollution months, ascribing to lower
contributions of OVOCs.

331        The VOCs concentration was mainly affected by local emissions on $PM_{2.5}$ polluted days according to the

PSCF result. Besides, the air masses that originated from Inner and passed through western Hebei also played an
important role in affecting the VOCs concentration as indicated by the relatively high VOCs concentration and
SOAFP for this trajectory cluster. The south trajectories originating from the junction of Hebei province exhibited
relatively low VOCs concentration. However, the SOAFP of this trajectory cluster was relatively high because
of high proportion. Specifically, the proportion of southwest trajectories with high-density emissions to the total
trajectories was higher on $PM_{2.5}$ pollution days (65.6%) than that on clean days (25.3%). During the low-$PM_{2.5}$
months, the VOCs concentration was mainly affected by the short distance northern (42.9%, cluster 2 in Fig. 11c)
and western trajectories (29.8%, cluster 1 in Fig. 11c). Although the proportion of the southern trajectories
(cluster 4 in Fig. 11c) to the total trajectories was relatively lower, both the VOCs concentration and the SOAFP
of this trajectory cluster was the highest. Overall, the above results indicated that the transmission of highly
polluting air masses from the junction of western and southern Hebei province should be paid attention for
controlling emissions of reactive VOCs compounds and alleviating $PM_{2.5}$ pollution.
**4. Conclusions**
In this work, the field sampling campaign of VOCs was conducted during December 2018 and November 2019
to investigate the characteristics, sources and secondary transformation potential the role of VOCs at an urban
site in Beijing. In total, 95 VOCs including 25 alkanes, 8 alkenes, 16 aromatics, 34 halocarbons and 12 OVOC



were identified and quantified. The VOCs concentrations during the sampling period ranged from 5.5 to 118.7
ppbv with mean value of 34.9 ppbv. In terms of the composition, alkanes, OVOCs and halocarbons were the
dominant chemical groups, accounting for 75-81% of the TVOCs across the sampling months. Nine $O_3$ pollution
events and fourteen $PM_{2.5}$ pollution events were observed during the sampling period. By excluding the
meteorological impact, the $O_3$ level driven by emissions during the $O_3$-polluted months were higher than that
during the $O_3$-clean months, and similar pattern was found for $PM_{2.5}$. The molar ratio of VOCs to $NOx$ indicated
that $O_3$ formation was limited by VOCs during both the $O_3$-polluted and $O_3$-clean months, and thus reducing
VOCs emission is essential for alleviation of $O_3$ pollution.
Six VOC sources were identified based on the PMF including coal/biomass combustion, solvent use,
industrial sources, oil/gas evaporation, gasoline exhaust and diesel exhaust. By considering both the
concentration and maximum incremental reactivity of individual VOC species for each source, fuel use and diesel
exhaust sources particularly toluene, xylenes, trans-2-butene, acrolein, methyl methacrylate, vinyl acetate, 1-
butene and 1-hexene were identified as the main contributors of $O_3$ formation during the $O_3$-polluted months,
illustrating the necessity of conducting emission controls on these pollution sources and species for alleviating
$O_3$ pollution. VOCs from diesel vehicles and combustion were found to be the dominant contributors for SOAFP,
particularly the VOC species of toluene, 1-hexene, xylenes, ethylbenzene and styrene, and top priority should be
given to these for the alleviation of haze pollution. The PSCF analysis showed that $O_3$ and $PM_{2.5}$ pollution was
mainly affected by local emissions. Besides, the transmission of highly polluting air masses from the western and
southern Hebei should also be paid attention for $O_3$ and $PM_{2.5}$ pollution control.
**Acknowledgements**





This work was supported by the National Natural Science Foundation of China (21625701) and the Beijing
Municipal Science and Technology Project (Z191100009119001 & Z181100005418018).

**Data availability**

The meteorological data are available at http: //data.cma.cn/ (China Meteorological Administration). The
website can be browsed in English http://data.cma.cn/en. The concentrations of air pollutants including $PM_{2.5}$,
$O_3$ and $NO_X$ are available at https://air.cnemc.cn:18007/ (Ministry of Ecology and Environment the People's
Republic of China). The website can be browsed in English http://english.mee.gov.cn/.

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



**Figure captions**


**Figure 1.** Time series of meteorological parameters and levels of air pollutants during the sampling
period.
**Figure 2.** Comparison of the concentration and composition of major chemical groups observed in
2019 (this study), 2016 (Liu et al., 2020) and 2014 (Li et al., 2015).
**Figure 3.** Comparison of major meteorological parameters and air pollutants on clean and polluted
days.
**Figure 4.** Statistic decomposition of meteorological and emission contribution to $O_3$ and $PM_{2.5}$
levels during different periods.
**Figure 5.** OFP and SOAFP by chemical groups during different periods.
**Figure 6.** Source profiles of VOCs identified using the PMF model and the relative contributions of
the individual VOC species.
**Figure 7.** Contributions of each source to VOCs during different periods.
**Figure 8.** Contributions of each source to OFP and SOAFP during different periods.
**Figure 9.** OFP values of the dominant VOC species in the different source categories during the on
polluted (a) and clean (b) days of the high-$O_3$ months, and SOAFP values on polluted (c) and clean
(d) days of the high-$PM_{2.5}$ months.
**Figure 10.** Backward trajectory cluster analysis (24 h) and PSCF analysis during different periods:
(a) polluted days of the high-$O_3$ months, (b) clean days of the high-$O_3$ months, (c) low-$O_3$ months.
**Figure 11.** Backward trajectory cluster analysis (24 h) and PSCF analysis during different periods:
(a) polluted days of the high-$PM_{2.5}$ months, (b) clean days of the high- $PM_{2.5}$ months, (c) low- $PM_{2.5}$
months.




**Fig. 1.**

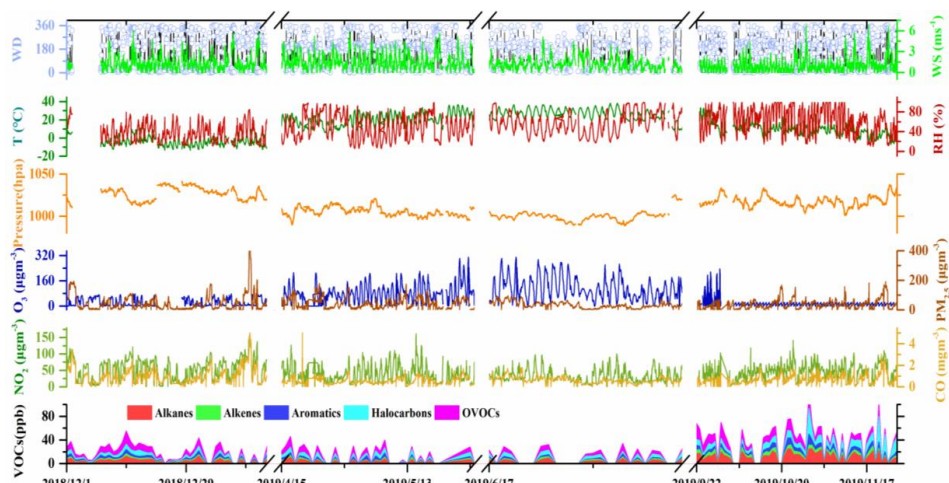








**Figure 2.**

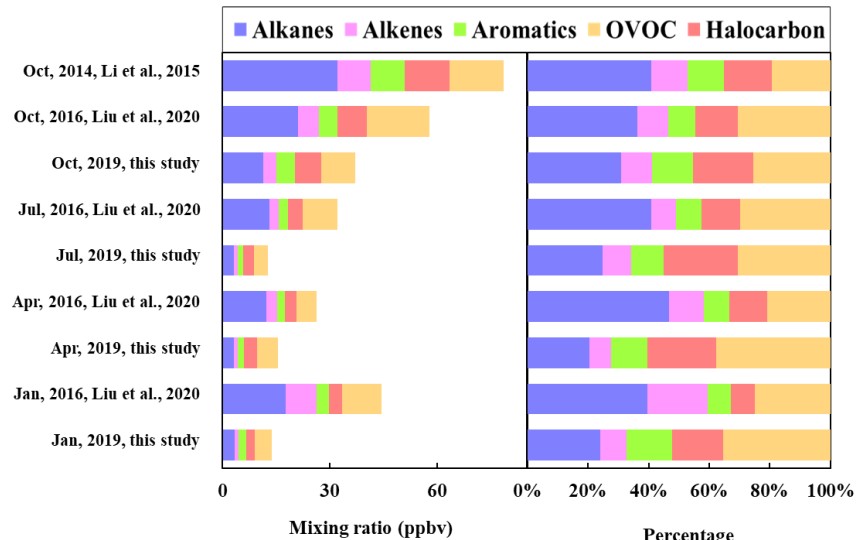






**Fig. 3.**

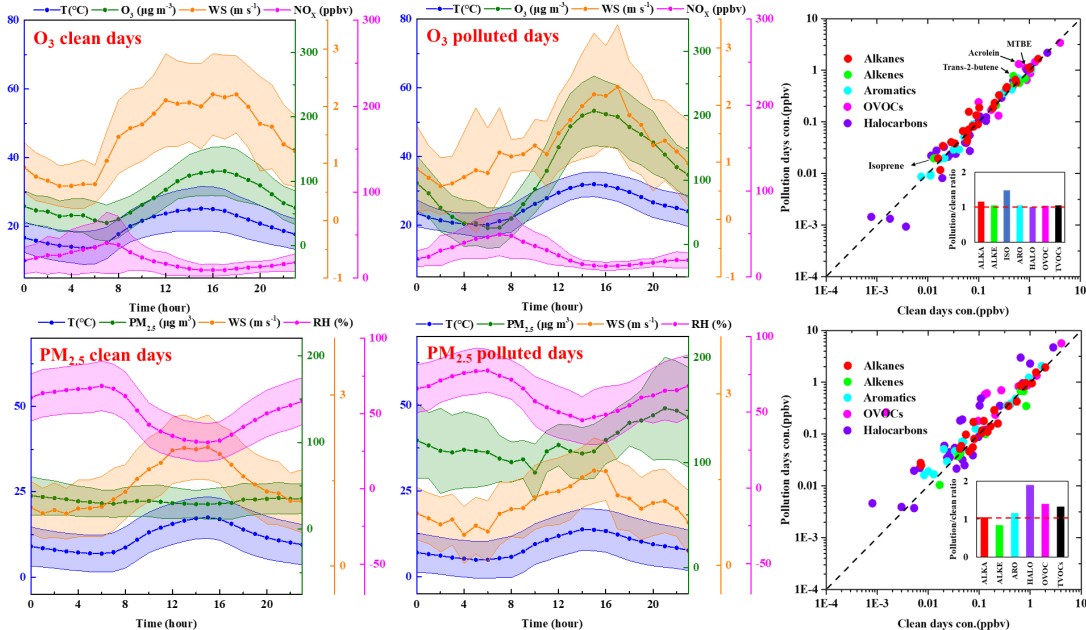






**Fig. 4**

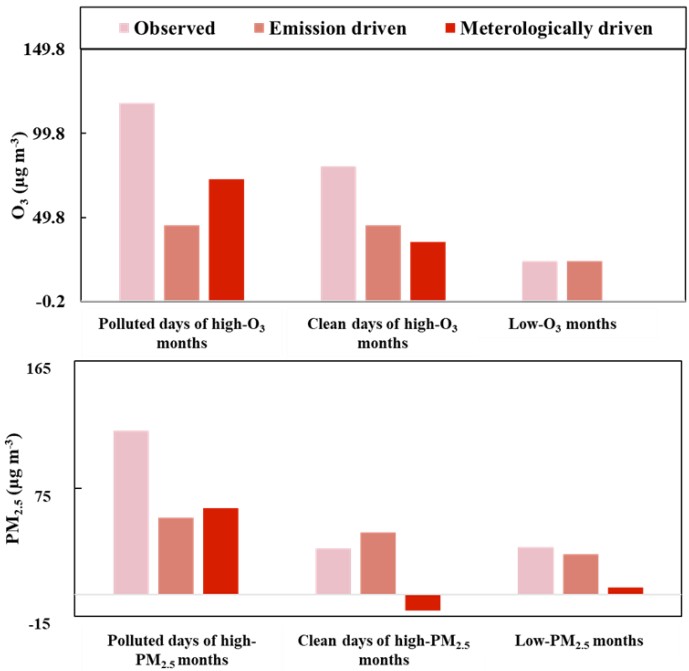






**Fig. 5**

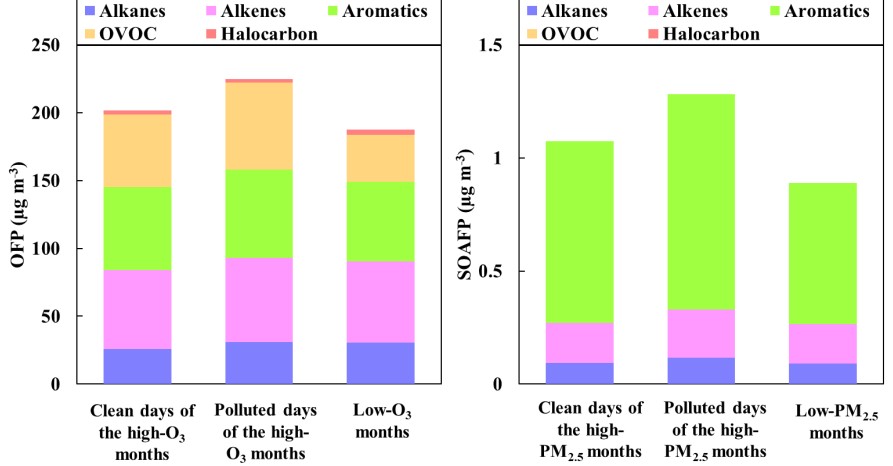







**Fig. 6.**



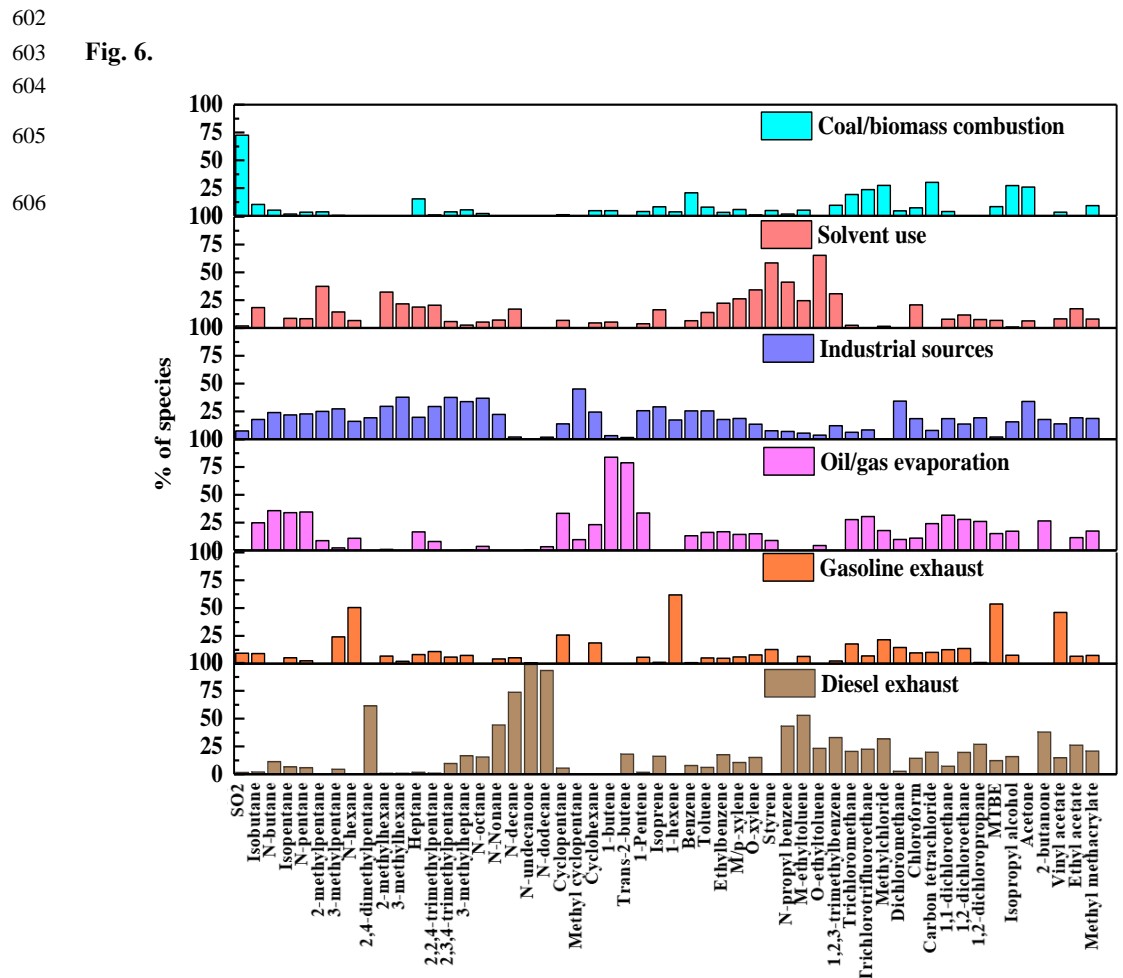





**Fig. 7**

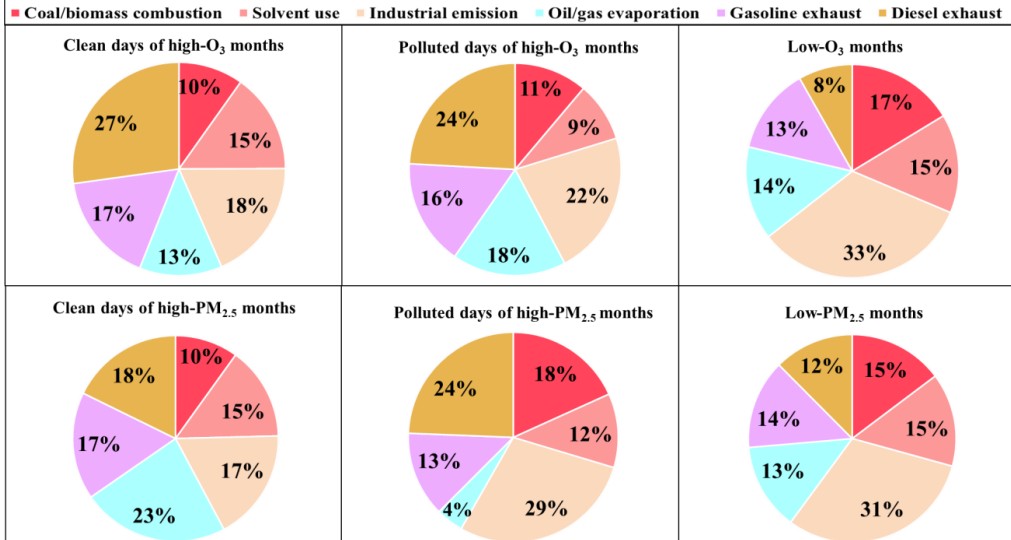




**Fig. 8**

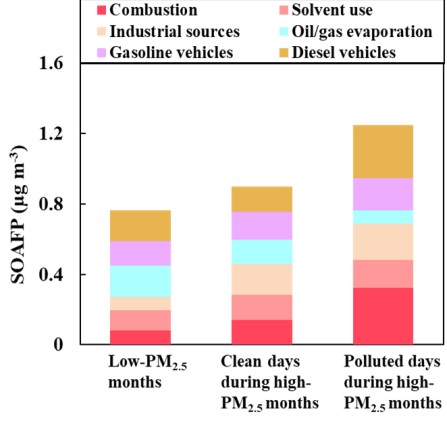
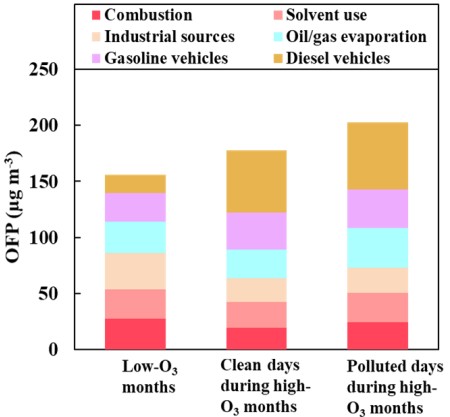






**Fig. 9**

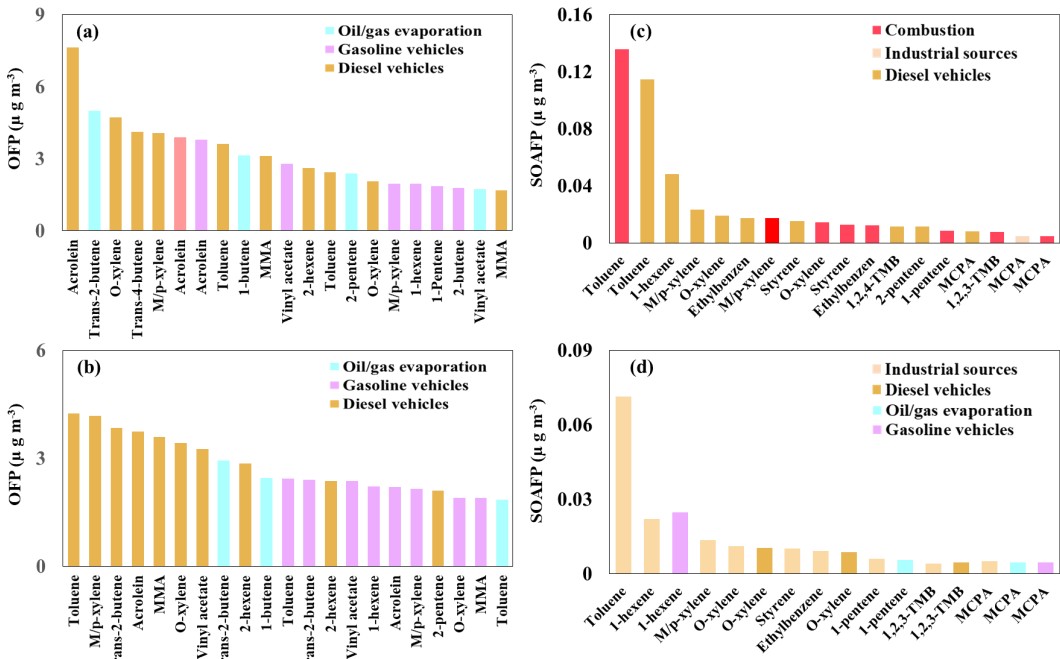




**Fig. 10**

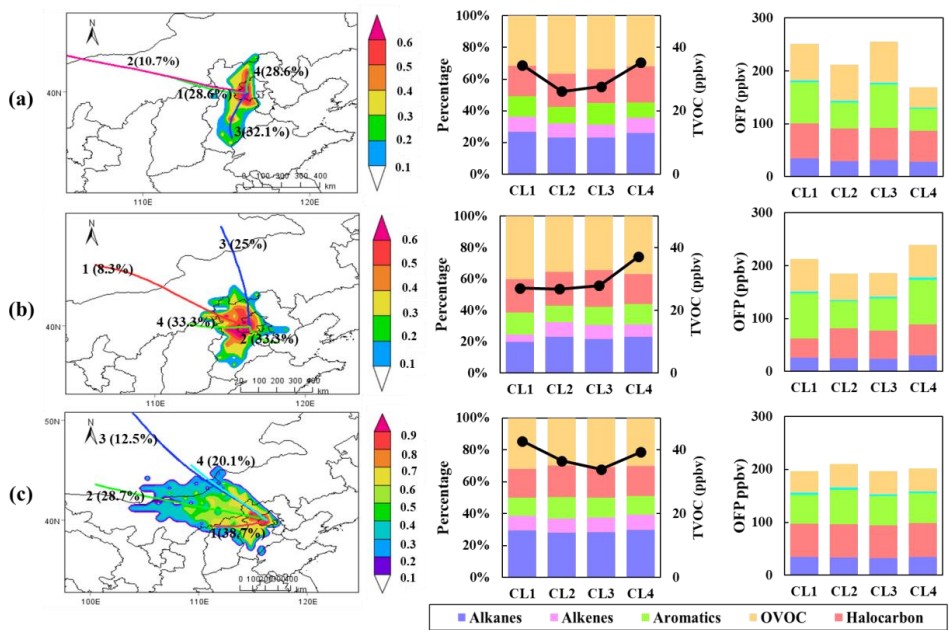




**Fig. 11**

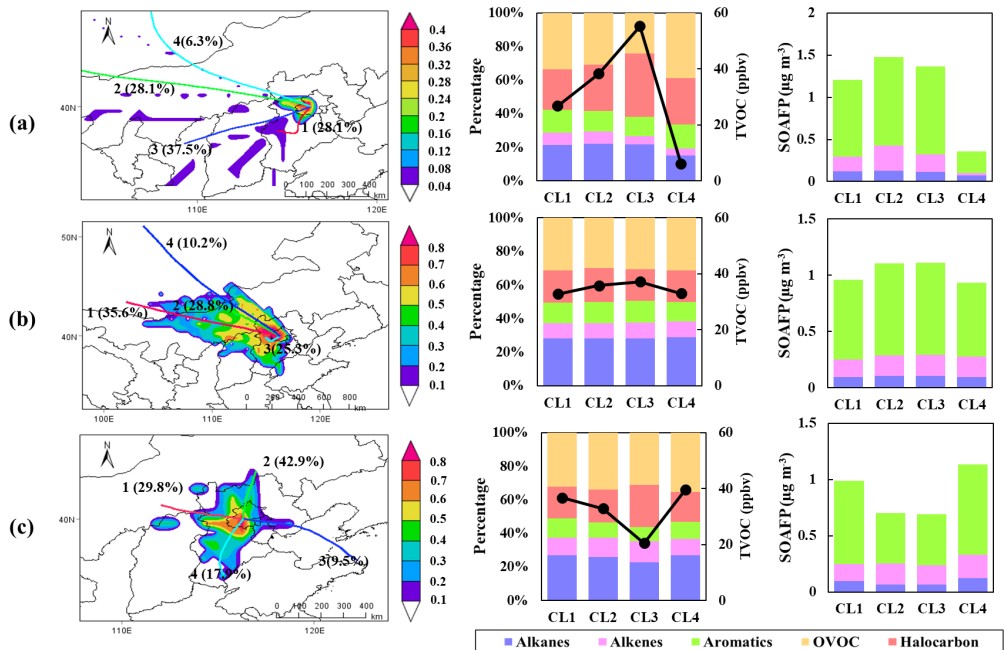
