# Peer review of "Measurement report: Ambient volatile organic compounds (VOCs) pollution at urban"

_Atmospheric Chemistry and Physics, 2021_

## Author Comment (AC1)

**Dear editor:**

Here we submit our revised manuscript for consideration to be published on **Atmospheric Chemistry and Physics**. The further information about our manuscript is as follows:

**Topic:** Measurement report: Ambient volatile organic compounds (VOCs) pollution at urban Beijing: characteristics, sources, and implications for pollution control

**Type of Manuscript:** article

**Authors:** Lulu Cui[1], Di Wu[1], Shuxiao Wang[1,2*], Qingcheng Xu[1], Ruolan Hu[1], Jiming Hao[1, 2]

**Corresponding author:**

**Shuxiao Wang.** Address: School of environment, Tsinghua University, Beijing 100084, China; Tel.: (+86)20- 62771466; E-mail: shxwang@tsinghua.edu.cn.

We acknowledge the suggestions of the editor, and are also grateful to your efficient serving. We have updated the manuscript on the basis of these valuable comments. Our responses were listed as following:

**Reviewer 1:** Cui et al. conducted a campaign of comprehensive field observations at an urban site in Beijing. The composition, sources, and secondary transformation potential of VOCs were also identified. Overall, the study is very interesting and shows some new findings. However, the manuscript still suffers from many flaws especially the language expression. Furthermore, section 3.2.1 is not well-organized and needs major revisions. The detailed comments are as follows:

**Comment 1:** Why not perform the hourly measurement of VOCs? To the best of my knowledge, the daily resolution for VOCs measurement is too coarse. Especially, the PMF model needs substantial observation data, which ensures the model's reliability.

**Response:** In this work, the VOCs samples were collected using the summa canisters and then measured by the GC-MS technique. Although on-line monitoring techniques such as PTR-TOF-MS show high time resolution (e.g., a few minutes), they cannot detect certain hydrocarbons such as alkanes which are widely in air. Therefore, we chose to measure the VOCs concentrations using the GC-MS technique. In this work, the air samples were collected during December 2018 and November 2019 with a total of 123 effective sampling days. Three groups of samples were collected each day, i.e., daytime samples, nighttime samples and all-day samples. Therefore, 369 sets of data were obtained, which is enough for PMF analysis. The PMF model showed the lowest $Q$ (robust) and $Q$ (true) values, and thus the result of PMF model is robust.

**Comment 2:** The authors need to add the detailed QA/QC of VOCs and other criteria pollutants in section 2.1. The information is very important otherwise the study might be meaningless.

**Response (Lines 110-114, 121-122):** Thanks for the reviewer's suggestion. The QA/QC of VOCs has been described in the manuscript, as shown in lines 110-114:

"Quality assurance and quality control, including method detection limit (MDL) of each compound, laboratory and field blanks, retention time, accuracy and duplicate

measurements of samples were performed according to USEPA Compendium Method TO-15 (USEPA 1999). The correlated coefficients of the calibration curves for all the compounds were > 0.95. The relative standard deviation (RSD) for all of compounds of triplicates were 0.5%-6.0%."

The other pollutants including $PM_{2.5}$, $NO_X$ and $O_3$ were analyzed using oscillating balance analyzer (TH-2000Z, China), the $NO$–$NO_2$-$NO_X$ Analyzer (Thermo Fisher Scientific USA, 17I), and the Ozone Analyzer (Thermo Fisher Scientific USA, 49I), respectively. The quality assurance of $NO_2$, $O_3$, and $PM_{2.5}$ was conducted based on HJ 630-2011 specifications.

**Comment 3:** Section 2.3: Why do you use RF model rather than other decision tree model or chemical transport model (CTM)? The predictive performance of RF model might be worse than GBDT and XGBoost. Meanwhile, CTM is a process-based model, which could clearly explain the contribution of many VOC species to $O_3$ Moreover, the hyperparameter of RF model should be added.

**Response:** The impact of meteorology and emission on $PM_{2.5}$ and $O_3$ can be separated by both statistical methods and CTMs. Although CTMs have the advantage of simulating the atmospheric processes of pollutants, these model simulations require considerable computation resources. Besides, the uncertainties of model inputs (emission inventory and meteorology) will lead to simulation deviation, and researchers' selection of chemical reaction mechanisms as well as parameter optimization leads to varying results (Chen et al., 2020). Increasing more studies use statistic methods (e.g.,

multiple linear regression, GAM and machine-learning models, etc.) to separate the contributions of meteorology and emission to air pollution variations (Xiao et al., 2021; Grange et al., 2018; Vu et al., 2019; Zhang et al., 2020; Qu et al., 2020). RF model is a typical machine learning method and has shown good performance in separating the impacts of meteorology and emission on air pollutants (Li et al., 2021). In the present study, the coefficients of determination ($R^2$) for the RF model in predicting $PM_{2.5}$ and $O_3$ are 0.85 and 0.91, respectively, suggesting that the RF model is reliable.

**Comment 4:** Section 2.4: The BS, DISP, and BS-DISP tests should be also added.

**Response (Lines 162-167):** Thanks for the reviewer's suggestion. The BS, DISP, and BS-DISP tests of the PMF analysis has been added in the revised version:

"During the PMF analysis, the bootstraps (BS) method, displacement (DISP) analysis, and the combination of the DISP and BS (BS–DISP) were used to evaluate the uncertainty of the base run solution. A total of 100 bootstrap runs were performed, and acceptable results were gained for all factors (above 90%). Based on the DISP analysis, the observed drop in the $Q$ value was below 0.1 %, and no factor swap occurred, confirming that the solution was stable. The BS–DISP analysis showed that the observed drop in the $Q$ value was less than 0.5 %, demonstrating that the solution was useful."

**Comment 5:** Section 2.5: I think the PSCF analysis is not important in this study and could be removed.

**Response:** The reviewer's suggestion is reasonable. The PSCF analysis has been deleted in the revised version.

**Comment 6:** Section 3.2.1: Why not distinguish the meteorological and emission contributions to each VOC species?

**Response:** We tried to distinguish the meteorological and emission contributions to each VOC species by both RF and GBDT models. The results showed that the two models showed poor performance for predicting aromatics, halocarbons and OVOC species, with $R^2 < 0.4$ for most species, and we have not found the optimal proxy for predicting these species. In future studies, we hope add more useful variables to elevate the modelling performance of these VOC species.

**Comment 7:** Section 3.3: The source identification method of each source based on VOC fingerprint should be added in this part. I think this part is too rough and should be rewritten.

**Response (Lines 297-319):** Thanks for the reviewer's suggestion. The source identification method of each source based on VOC fingerprint has been added in the revised version as below.

**"3.3.1 Indication from tracers**

[revised manuscript text omitted]

**Comment 9:** Data availability: I suggest the authors open the VOC dataset and it is very valuable to some researchers engaged in air quality modelling.

**Response:** The VOC dataset will be upload to the Supplementary Information.

**Comment 10:** The English throughout the manuscript should be significantly revised.

**Response:** Thanks for the reviewer's suggestion. In the revised version, the English throughout the manuscript has been checked carefully.

**Reviewer 2:** The research focused on ambient volatile organic compounds pollution at urban Beijing, and analyzed their characteristics, sources, and control effects. This study is of interest to the atmospheric scientists and suitable for the ACP. The observation data were detailed presented, the chemical composition and emission sources were analyzed aiming at different months and different O3- or PM2.5- pollution days, and the VOCs decline was found through comparing with reference results to

support the control effects. However, I have a few concerns that should be addressed before the acceptance of the manuscript.

**Major comments:**

**Comment 1:** In introduction section, the air pollution status has greatly changed in past several years in Beijing, due to the strict control measures implemented. However, the corresponding introductions were outdated and can't present the current pollution characteristics. For example, line42 about SOA fraction in $PM_{2.5}$, line 48 about SOA contribution to haze pollution, line 56 about the contribution of biogenic and anthropogenic sources, and so on. The recent references and their conclusions should be referred to.

**Response (Lines 41-45,49-51, 53-55):** Thanks for the reviewer's suggestion. The recent references and their conclusions have been added in the revised version is as below.

Lines 41-45: "Besides, haze pollution occurred in urban sites in recent years were commonly characterized by enhanced formation of secondary organic aerosols (SOA) in fine particles, e.g., the fraction of SOA in organic aerosols has reached 58% in Xi'an during winter 2018, and 53% in urban Beijing during winter 2014 (Kuang et al., 2020; Li et al., 2017b; Sun et al., 2020; Xu et al., 2019)."

Lines 49-51: "Besides, the VOCs compounds including aromatics and biogenic species have significant impact on SOA formation which play an important role in haze formation (Huang et al., 2014; Tong et al., 2021)."

Lines 53-55: "VOCs in ambient air can be emitted by a variety of sources including both anthropogenic and biogenic sources. While biogenic emissions are significantly greater than anthropogenic emissions globally (Doumbia et al., 2021; Sindelarova et al., 2022), "

**Comment 2:** Methodology section, VOCs detection system should be GC-MS, but not GC (as mentioned in lines 95-96), for Agilent 5975 uses mass spectrometry detector. If the detector only included MSD but not included FID, C2 hydrocarbons would not be detected but they widely exist in atmosphere. This point should be illustrated. In addition, the efficiency of this analyze system for aldehydes should be well discussed. Because various monitoring standards don't explicitly recommend the "canister sampling-GC/MS analyzer" to detect aldehydes.

**Response (105-107, 114-116):** Thanks for the reviewer's suggestion. In the revised version, the point that GC-MS cannot detect VOCs compounds (C2-C3) with low boiling point (i.e., ethane, ethene, acetylene, and propane) has been illustrated. The efficiency of the GC-MS system for aldehydes was also discussed in the revised version.

**Comment 3:** This study used the fact that $O_3$ or $PM_{2.5}$ pollution event happening to define high-$O_3$ months (Apr, May, Jun, Jul and Sep) and high-$PM_{2.5}$ months (April,

May, Oct, Nov, Dec, Jan). It seems weird. For example, although $O_3$ event never happened in Aug, but ozone level was also relatively higher in Aug than in Apr and Sep. So Aug should be considered as the high-$O_3$ month, comparing with Apr and Sep. And then, in the results of PMF, the source apportionment in low-$O_3$ months (Oct, Nov, Dec, Jan) was different with that in high-$PM_{2.5}$ months (April, May, Oct, Nov, Dec, Jan), but similar to that in low-$PM_{2.5}$ months (Jun, Jul and Aug). This conclusion was unreasonable to a certain extent.

**Response:** We agree with the reviewer's suggestion. In the revised version, the months with $O_3$ pollution events (days with maximum 8-h average $O_3$ exceeding 160 µg m$^{-3}$) were defined as the $O_3$-polluted months, and with $PM_{2.5}$ pollution events (daily average $PM_{2.5}$ exceeding 75 µg m$^{-3}$) were defined as the $PM_{2.5}$-polluted months.

In the initial version, the source appointment during the "low-$PM_{2.5}$ months" was confused with that on "clean days of high-$PM_{2.5}$ months". In the revised version, the error has been corrected in both the main text and Figure 8.

**Comment 4:** When using PSCF to explore the spatial potential sources of VOCs in urban Beijing, 24h was considered for all species. However, the lifetimes of various VOCs species were greatly different, several hours for alkenes, but several days for some alkanes and halocarbons. I suggest various groups of VOCs should be individually considered, to give the lifetime hours in backward trajectories.

**Response:** Thanks for the reviewer's suggestion. According to the comments of the first reviewer, the PSCF analysis is not important and could be removed. In the revised version, the contents about the PSCF analysis have been deleted.

**Minor comments:**

**Comment 1:** Abstract: "$O_3$/$PM_{2.5}$" frequently appeared but without an explicit definition. It is hard to understand the "high and low-$O_3$/$PM_{2.5}$ months", "$O_3$/$PM_{2.5}$ polluted days", and "high $O_3$/$PM_{2.5}$ levels", etc.

**Response:** Thanks for the reviewer's suggestion. In the revised version, the different pollution periods have been clearly defined, i.e., the months with $O_3$ and $PM_{2.5}$ pollution events were defined as the $O_3$-polluted and $PM_{2.5}$-poluted months, whereas the months without $O_3$ and $PM_{2.5}$ pollution events were defined as the non-$O_3$-polluted and non-$PM_{2.5}$-poluted months, respectively. During the $O_3$-polluted months, the days with maximum 8-h average $O_3$ exceeding 160 µg m$^{-3}$ and below 160 µg m$^{-3}$ were defined as the $O_3$ pollution days of the $O_3$-polluted months and $O_3$ compliance days of the $O_3$-polluted months, respectively. During the $PM_{2.5}$-polluted months, the days with average $PM_{2.5}$ exceeding 75 µg m$^{-3}$ and below 75 µg m$^{-3}$ were defined as the $PM_{2.5}$ pollution days of the $PM_{2.5}$-polluted months and $PM_{2.5}$ compliance days of the $PM_{2.5}$-polluted months, respectively.

**Comment 2:** Lines 32-34: "The positive matrix factorization (PSCF) analysis showed that $O_3$ and $PM_{2.5}$ pollution was mainly affected by local emissions." PSCF was conducted for VOCs, but not for ozone and $PM_{2.5}$. No evidence to support this conclusion.

**Response:** Thanks for the reviewer's suggestion. According to the first reviewer's suggestion, the contents about the PSCF analysis have been deleted in the revised version.

**Comment 3:** Line 47: VOCs chemistry in ozone formation involves gas-phase reaction, but not multiphase reaction.

**Response (Line 45-46):** Thanks for the reviewer's suggestion. The error has been corrected in the revised version.

**Comment 4:** Line 104: the "coefficient" should be coefficients; "was" should be "were"

**Response (Line 112-113):** Thanks for the reviewer' rigorous. The error has been corrected in the revised version.

**Comment 5:** Line 111-112: air pressure appeared twice.

**Response (Line 123):** It's our overlook. One of the "air pressure" has been deleted in the revised version.

**Comment 6:** More detailed model performance verifications (RF) are necessary, although $R^2$ has provided in Fig. S2.

**Response:** The reviewer's suggestion is reasonable. In the revised version, the RMSE and MAE of the RF model in predicting $PM_{2.5}$ and $O_3$ have been added in Fig. S2.

**Comment 7:** Line 191-193: I cannot figure out the sentence, suggesting checking out syntax rules.

**Response (213-215):** Thanks for the reviewer's suggestion. The sentence has been rewritten in the revised version.

**Comment 8:** Line 234: Fig. S3 was mentioned, however, there is not Fig. S3 in the supplement of this passage.

**Response:** Thanks for the reviewer's suggestion. The Fig. S3 has been added in the supporting information.

**Comment 9:** line 243-244: "Alkenes, aromatics and OVOCs were the three contributing chemical groups to O3 formation", should be "the three biggest contributors".

**Response (269):** Thanks for the reviewer's suggestion. This sentence has been revised.

---

## Author Response (AR2)

**Dear editor:**

Here we submit our revised manuscript for consideration to be published on **Atmospheric Chemistry and Physics**. The further information about our manuscript is as follows:

**Topic:** Measurement report: Ambient volatile organic compounds (VOCs) pollution at urban Beijing: characteristics, sources, and implications for pollution control

**Type of Manuscript:** article

**Authors:** Lulu Cui[1], Di Wu[1], Shuxiao Wang[1,2*], Qingcheng Xu[1], Ruolan Hu[1], Jiming Hao[1, 2]

**Corresponding author:**

**Shuxiao Wang.** Address: School of environment, Tsinghua University, Beijing 100084, China; Tel.: (+86)20- 62771466; E-mail: shxwang@tsinghua.edu.cn.

We acknowledge the suggestions of the editor, and are also grateful to your efficient serving. We have updated the manuscript on the basis of these valuable comments. Our responses were listed as following:

**Main Comments**

**Comment 1:** From the abstract and text (Sect. 2.1), it is not made clear to the reader that these are not continuous measurements. For instance, there are no measurements from mid-January to mid-April or from mid-July to mid-September. This point should be made more transparently in the manuscript. Additionally, the potential limitations/biases of the analysis, particularly for the "ozone polluted months" given

these measurement limitations should be briefly discussed.

**Response (Lines 11-16, 74-76, 344-351):** Thanks for the editor's suggestion. In the revised version, the observation periods have been clearly described in the abstract section as below:

"In this work, four intensive field measurements of VOCs during winter of 2018 (from 1 December of 2018 to 17 January of 2019), spring (15 April to 27 May), summer (17 June to 13 July) and autumn (22 September to 27 November) of 2019 were conducted at an urban site in Beijing to characterize VOCs sources and their contributions to air pollution."

The observation periods have also been clearly described in the last paragraph of the introduction section as below:

"In this work, ambient air samples were collected at an urban site in Beijing from December 2018 to mid-January 2019, mid-April to late May 2019, mid-June to mid-July 2019, and late September to late November 2019, respectively".

The potential limitations of the analysis due to the measurement limitations have been added in the revised version as below:

"3.4 Limitation

This study analyzed the VOC sources and their contributions to $O_3$ and SOA formation across different seasons. It should be pointed out that the sampling campaign for VOCs measurement was not conducted continuously during December 2018 and November 2019. For instance, the air samples were not collected in August and February-March of 2019, during which the pollution events of $O_3$ and $PM_{2.5}$ occurred, respectively. The

variations, sources and secondary transformation potentials of VOCs, particularly for $O_3$ and $PM_{2.5}$ pollution periods cannot be fully depicted. Despite the uncertainties that remained, the results obtained in this study provide useful information for VOCs emission control strategy and assist overcoming air pollution issue in Beijing."

**Comment 2:** More description about the RF model needs to be included in the manuscript (SI ok). For instance, how were the meteorological predictors included given the long sampling time for the VOCs and strong diel profiles of many of the meteorological variables? Further details about the training dataset would also be useful. For instance, is it representative across season, meteorological variables, weekday vs weekend, etc. or are some time periods overrepresented in the training vs test dataset? This is important given the relatively small number of time points.

**Response (Lines 141, 145-146):** Thanks for the editor's suggestion. In this study, the RF model relates the hourly variability of $O_3$ and $PM_{2.5}$ to that of meteorological variables. The sentence "The modelling relates the hourly variability of $O_3$ and $PM_{2.5}$ to that of meteorological variables." has been added in the revised version.

For the 10-fold CV approach, the training dataset in each round includes ~90% randomly selected data representing different seasons. In the revised version, the sentence "In each round, the training dataset includes ~90% randomly selected data representing different seasons" has been added.

**Comment 3:** Thank you for including the spreadsheet with the measurements. This is an important improvement in data availability. I think the data availability and the impact of the manuscript would be improved with some modifications. In particular, it

may be worth reporting the measurements in a more standardized format (for instance ICARTT format). At a minimum, please consider the following suggestions

a. Please standardize the names of the chemical compounds throughout the manuscript, the spreadsheet, and Table S2. For instance, the manuscript lists chloromethane whereas in the spreadsheet it is listed as methyl chloride. Including a CAS number for each compound would be helpful.

**Response:** The names of the chemical compounds in the text, Table S1, and Table S2 have been unified. Besides, the CAS number for each compound have been added in Table S1 and Table S2.

b. Is there a difference between the files "Table S1 – VOCs mixing ratios" and "Table S1 – VOCs mixing ratios (ppbv)"? Please include the units within the spreadsheet and not just in the file names.

**Response:** There is no difference between the files "Table S1 – VOCs mixing ratios" and "Table S1 – VOCs mixing ratios (ppbv)". The units within the spreadsheet have been added in the file names.

c. The mixing ratios reported in the files should have the correct number of significant figures.

**Response:** The mixing ratios of the VOCs species in Table S1 have been adjusted to two significant digits after the decimal point.

d. Please consider including an indication of the uncertainty and detection limit for each VOC in the spreadsheet. Please also consider indicating when a species was below detection limit with a standardized value (e.g., as per the ICARTT standard) rather than

reporting as zero.

**Response:** The uncertainty and detection limit for each VOC have been added in the spreadsheet. Besides, the VOCs species with concentrations below the detection limit were marked as "BDL" in the spreadsheet.

e. Add mention of the spreadsheet to the data availability section.

**Response (376-377):** The mention of the spreadsheet has been added in the data availability section: "The daily mixing ratio of individual VOCs species is given in Table S1 in the Supplement".

**Technical**

**Comment 1:** Line 101: "suitability" is probably a better word choice than "availability"

**Response (Line 95):** Thanks for the editor's suggestion. The word "availability" has been replaced by "suitability" in the revised version.

**Comment 2:** Lines 102-104: The description of the temperatures is confusing to me. Both -40 °C and 90 °C are listed as initial temperatures. I assume they are referring to different points, but it is not clear. Additionally, is the temperature ramped to 220 °C? If so, please include the rate.

**Response (97-99):** We are sorry for the wrong expression in the manuscript. The description of the oven temperature has been rewritten in the revised version: "The oven temperature was programmed at 40 °C for 3 minutes initially, then raised to 90 °C at 8°C per minute, and later raised 220 °C at 6°C per minute, holding for 9 minutes."

**Comment 3:** Lines 116: Acrolein is listed in the SI and excel spreadsheet, but

ethylacrolein is not. Should this be a reference to acrolein?

**Response (Line 111):** It's our overlook. The word "ethylacrolein" has been corrected into "acrolein" in the revised version.

**Comment 4:** Line 122: A word is missing after "meteorological." Variables could be an acceptable choice.

**Response (Line 119):** The word "variables" has been added after "meteorological".

**Comment 5:** Line 157: How was $SO_2$ measured? $SO_2$ measurements were not mentioned in Sect. 2.1.

**Response (Lines 114 and 119):** $SO_2$ was measured by a pulsed UV fluorescence (Thermo 43i, USA) with the detection limit of 0.5 ppbv. The description of $SO_2$ measurement has been added in the revised version.

**Comment 6:** Line 195/Figure 2: Please add a short sentence on the overlap of VOCs measured between this study and the ones shown in Figure 2 so that the reader can quickly judge that this trend is real and not an artifact of measuring different subsets of VOCs.

**Response (Line 183):** Thanks for the editor's suggestion. The overlap of VOCs groups measured between this study and the ones shown in Figure 2 has been described in the revised version as below:

"As shown in Fig. 2, the concentrations of TVOCs and major VOC groups including alkanes, alkenes, aromatics, halocarbons and OVOCs observed in this study were apparently lower than those during the sampling months in 2014 and 2016 in urban Beijing (An et al., 2012; Liu et al., 2020a; Li et al., 2015b), indicating the effectiveness

of control measures in most recent years on lowering VOCs emission.".

**Comment 7:** Lines 201-202: The dates listed for the ozone pollution days cover more than 14 days. Please clarify.

**Response (Lines 190-191):** We are sorry for the mistake. The date of 14 $O_3$ pollution days have been corrected in the revised version.

**Comment 8:** Line 207: Five, not four" ozone polluted months are listed earlier (line 203). Please clarify.

**Response (Line 194):** We are sorry for the mistake. The error has been corrected in the revised version.

**Comment 9:** Line 228: Typo. Should be "April"

**Response (Line 216):** We are sorry for the mistake. The error has been corrected in the revised version.

**Comment 10:** Line 235: I am not convinced that a correlation between RH and TVOCs really suggests that the "secondary transformation of VOCs was more conducive at higher RH." There could be co-varying factors that are not explored here. I suggest to either support this with more analysis or remove it.

**Response (219-220):** Thanks for the reviewer's suggestion. The sentence "Both the value of relative humidity (RH) and TVOCs increased significantly on $PM_{2.5}$ pollution days, suggesting that the secondary transformation of VOCs was more conducive at higher RH." has been deleted in the revised version.

**Comment 11:** Line 273: Please consider reversing the ordering of compliance and polluted days here since the reverse ordering was used earlier. It would make it easier

to read.

**Response (Lines 256-260):** The ordering of compliance and polluted days here has been reversed in the revised version.

**Comment 12:** Lines 298-300: I think this sentence is oversimplifying a complex situation. VOC mixing ratios can also exhibit "great change" due to meteorological factors (e.g., shallow boundary layers, low transport, etc.). I suggest focusing this first sentence on ratios rather than on absolute abundances.

**Response (Lines 280-282):** Thanks for the editor's suggestion. The sentence has been corrected into "The great changes in mixing ratios of different species are mainly affected by the photochemical processing and the emission inputs, and the ratios of VOCs species having similar atmospheric lifetimes can reflect the source features".

**Comment 13:** Line 403: I suggest changing the wording to "…from December 2018 to November 2019…"

**Response (Lines 353-354):** Thanks for the editor's suggestion. The revision has been made.

**Comment 14:** Figures: Please add letters to identify the subpanels to all figures and include this naming in the manuscript and figure captions.

**Response:** Letters for identifying the subpanels to all figures, and the naming in the manuscript and figure captions have been added in the revised version.

**Comment 15:** Figure 2: It would be easier for the reader to understand this figure if horizontal dashed lines (or some other separator) were included to separate out the different months (e.g., b/w Oct 2019 and July 2016, etc.).

**Response:** In the revised version, horizontal dashed lines have been in Figure 2 to separate out the different months.

**Comment 16:** Figure 3: Please clarify the caption with regards to which days are represented in the figures. For instance, do the ozone compliance and ozone pollution days only include the ozone polluted months or is all the data included? Please consider using the same colors for the VOCs as is used in other figures. For instance, alkenes are green here but pink in figure 5.

**Response (Lines 660-667):** The caption with regards to which days are presented in Figure 3 has been added in the revised version. Besides, the colors for the VOCs in Figure 3 and Figure 5 have been unified.

**Comment 17:** Figure 4: Why does the y-axis on the top panel go to -0.2?

**Response:** According to our results, the meteorological conditions are generally favorable for $O_3$ decrease during the non-$O_3$-polluted months, and the mean meterologically-driven $O_3$ concentration during the non-$O_3$-pollted months was -0.19 $\mu g\ m^{-3}$.

**Comment 18:** Figure 9: Please consider structuring Figures 5 and 9 with the same format rather than reversing the ordering.

**Response:** Figure 5 and 9 have been structured with the same format in the revised version.